# Effects of Dietary Fiber Sources during Gestation on Stress Status, Abnormal Behaviors and Reproductive Performance of Sows

**DOI:** 10.3390/ani10010141

**Published:** 2020-01-15

**Authors:** Shuangbo Huang, Jianfu Wei, Haoyuan Yu, Xiangyu Hao, Jianjun Zuo, Chengquan Tan, Jinping Deng

**Affiliations:** 1Guangdong Provincial Key Laboratory of Animal Nutrition Control, Institute of Subtropical Animal Nutrition and Feed, College of Animal Science, South China Agricultural University, Guangzhou 510642, Guangdong, China; shuangbohuang@sina.com (S.H.); yuhy02@sina.com (H.Y.); xiangyvhao@sina.com (X.H.); zuoj@scau.edu.cn (J.Z.); 2Guangzhou DaBeiNong Agri-animal Huabandry Science and Technology Co., Ltd., Guangzhou 510642, Guangdong, China; weijianfu@dbn.com.cn; 3National Engineering Research Center for Breeding Swine Industry, College of Animal Science, South China Agricultural University, Guangzhou 510642, Guangdong, China; 4National Engineering Laboratory for Pollution Control and Waste Utilization in Livestock and Poultry Production, Institute of Subtropical Agriculture, Chinese Academy of Sciences, Changsha 410125, Hunan, China

**Keywords:** resistant starch, fermented soybean fiber, stillbirth rate, abnormal behaviors, stress, satiety, sows

## Abstract

**Simple Summary:**

This study aimed to appraise the effects of two available unconventional dietary fiber resources (resistant starch and fermented soybean fiber) on sows’ reproductive performance through an in vitro-in vivo method to facilitate their application in the rural livestock production systems. Results indicated that inclusion of 5% resistant starch with greater swelling capacity in the gestation diet was beneficial to enhancing postprandial satiety, alleviating stress status, reducing abnormal behaviors and thus lowering the stillbirth rate of sows.

**Abstract:**

Inclusion of fiber in gestation diets is a method for enhancing satiety and reducing abnormal behaviors in restricted feeding sows without providing excess energy. The purpose of this study was to use an in vitro-in vivo method to appraise the effects of two available unconventional dietary fiber resources during gestation on sows’ physio-chemical properties of diets, postprandial satiety, performance, abnormal behaviors, stress status and lactation feed intake under three different dietary treatments: control diet (CON diet), 5% resistant starch diet (RS diet), and 5% fermented soybean fiber diet (FSF diet) with a total of 78 (average parity 5) Landrace × Yorkshire sows. Results showed that swelling capacity was higher in the RS diet than in the CON or FSF diet. Meanwhile, the 48 h cumulative gas production and the final asymptotic gas volume after in vitro fermentation of gestation diets showed an increased trend (*p* = 0.07, *p* = 0.09, respectively) in the RS diet versus the CON or FSF diets. While the sows’ litter size, body weight, backfat or weaning-to-estrus interval were not affected (*p* > 0.05) by the three treatments during gestation, the RS group showed a decline in stillbirth number (*p* < 0.05) and stillbirth rate (*p* < 0.01) relative to the other two groups. Meanwhile, the proportion of standing was lower while the sow’s serum concentrations of PYY (peptide YY) and GLP-1 (glucagon-like peptide-1) were higher (*p* < 0.05) on day 70 of gestation in the RS group than in the CON or FSF group. Compared with the CON group, the RS group showed a downward tendency (*p* = 0.07) in the sows’ plasma cortisol concentration on day 70 of gestation. A comparison of oxidative and antioxidative indicators revealed an increase in the sows’ serum FRAP (ferric ion reducing antioxidant power) (*p* < 0.05) and a decrease of protein carbonyl (*p* < 0.05) on day 109 of gestation in the RS or FSF group versus the CON group. Overall, inclusion of 5% RS with greater swelling capacity in the gestation diet contributed to enhancing the postprandial satiety, alleviating the stress status, reducing the abnormal behaviors and thus lowering the stillbirth rate of sows.

## 1. Introduction

In actual production, restricted feeding during gestation is widely adopted to avoid excess weight gain, which is associated with farrowing and locomotion problems [1]. However, excessive restriction of feeding makes the sows less satisfied during pregnancy, leading to frequent occurrence of stereotypes, which will be accompanied by increases of stress and physical harm, and in turn a negative impact on their reproductive performance [2]. Previous studies have shown that females showing a high relative frequency of vacuum chewing during gestation produced fewer total pigs born and a decrease of embryo survival than those showing no vacuum chewing [2,3,4]. Inclusion of fiber in gestation diets is not only a method for promoting nutritional satiety and reducing apparent feeding motivation in sows without providing excess energy, but also has the potential to increase litter size [5,6].

Several previous studies have indicated that feeding high-fiber diets can influence the welfare and reproductive performance of gestating sows by decreasing stereotypical behavior [1] and increasing the number of piglets born [7]. Conversely, other studies have shown that dietary fiber had no effect on the number of stillbirths, piglets born and salivary cortisol concentrations [8,9]. These studies are inconsistent with each other about the response of feed to dietary fiber during pregnancy, which may depend, to a large extent, on the fiber source and type [10]. Thus, the effects of different dietary fibers during gestation on the performance of sows and their offspring remain highly controversial and await further clarification.

Starch, the major dietary source of carbohydrates, has been recently recognized with incomplete digestion and absorption in the small intestine as a normal phenomenon, leading to increasing research interest in nondigestible starch fractions [11,12]. These fractions, termed as “resistant starches (RS),” are shown by extensive studies to possess physiological functions like dietary fiber and are capable of escaping digestion in the small intestine. RS feature the desirable physicochemical properties such as swelling, viscosity increase, gel formation, and water-binding capacity, with useful applications in a variety of foods [13]. Specifically, RS allow the formation of low-bulk high-fiber products with improved texture, appearance, and mouth-feel properties (such as better organoleptic qualities) compared with traditional high-fiber products [14]. However, currently available feed tables on the chemical composition and nutritional values rarely take into consideration the RS level and its fractions, suggesting the necessity of the research on the availability of such starches [15,16]. Previous studies have shown that adding RS to diets during sow mixing could increase satiety and reduce sow aggression [17], but few reports are available on its positive effect on the physio-chemical properties of diet and the reproductive performance of sows.

Additionally, fermented soybean fiber is a highly active, high-quality and low-cost fiber source generated in the microbial fermentation processes. A previous study has shown that fermented soy fibers could significantly contribute to the hypolipidemic and antiinflammation effects of soybeans in vivo [18]. Thus far, to our best knowledge, no report has been published on the effect of dietary addition of fermented soybean fiber on the postprandial satiety, and reproductive and lactation performance of sows.

Therefore, the purpose of this study was to appraise the effects of two available unconventional dietary fiber resources on sows’ reproductive performance through an in vitro-in vivo method and to extend their applications in rural livestock production systems.

## 2. Materials and Methods

The experimental designs and procedures presented in this study were carried out in accordance with the Animal Care and Use Committee of the Institute of Subtropical Agriculture, Chinese Academy of Science. The ethical approval number is ISA-2019-025.

### 2.1. Animals, Diets and Housing

After breeding, a total of 78 multiparous (average parity 5) Landrace × Yorkshire sows were assigned to three different dietary treatments based on parity and body weight (BW). The three isoenergetic and isonitrogenous gestation diets were formulated based on corn, soybean meal and wheat bran: (1) control diet without supplementation (CON diet); (2) the second diet included 5% resistant starch (RS diet); (3) the third diet included 5% fermented soybean fiber (FSF diet). The diets were formulated to meet or exceed the nutrient requirements recommended by the National Research Council (NRC, 2012) [19]. The components of the three diets are shown in Table 1, and they had similar levels of digestible energy (DE) and crude protein (CP).

Sows were restricted to 2.0 kg of pregnancy diet per day on days 1 to 30 of gestation, 2.4 kg of pregnancy diet per day on days 31 to 85 of gestation, and 3.4 kg of pregnancy diet per day during late gestation. The diets were supplied twice a day (07:00 and 15:30 h) as restricted-feed from mating until the day of farrowing, and half of the daily feed was given at each meal. During the entire 21 d lactation period, the piglets had no access to the sow’s feed or to creep feed. Pregnant sows were given ad libitum access to water throughout the experiment. On the day of parturition, sows were fed 0.5 kg and the ration was gradually increased by 1.0 kg/d until the maximum ration was reached. Then the sows had free access to feed during the following days of lactation. On the morning of weaning (day 21 of lactation), the sows were deprived of feed and moved into gestation stalls.

### 2.2. Sample Collection

Before feeding and at 2 h after feeding on day 70 of gestation, and before feeding on day 109 of gestation, blood samples were collected into vacuum tubes (5 mL) and labeled heparinized tubes (5 mL) from 24 sows (eight sows per dietary treatment) by ear vein puncture with a minimum amount of stress. Collection tubes for blood samples of glucagon-like peptide-1 (GLP-1) and peptide YY (PYY) contained a mixture of inhibitors (dipeptidyl peptidase-4 inhibitor (10 μL/mL blood), aprotinin (50 μL/mL blood) and Pefabloc SC (50 μL/mL blood]) to prevent degradation of the peptides [20,21]. Samples for serum assays (tubes containing no anticoagulant) were stored at room temperature for 4 h and then centrifuged for 5 min at 5000 ×g at 4 °C to collect the serum, followed by storage in labeled microfuge tubes at −80 °C until further analysis. Plasma samples were obtained by centrifuging the blood samples in labeled heparinized tubes at 5000 ×g for 5 min at 4 °C and then stored at −80 °C for further analysis.

On day 70 of gestation, 24 fasted sows (eight sows per dietary treatment) were selected for saliva sampling. Saliva (3 mL) was sampled from each sow with saliva collection kit provided by Guangdong Dream Biotechnology Co., Ltd (Guangzhou, China) by fastening the cotton roll on a clamp and allowing the sow to chew on the cotton roll for approximately 30 s or until the cotton roll was saturated. The saliva samples collected were centrifuged for 20 min at 2000 ×g and 4 °C for separation of supernatant, which was stored at −80 °C until analysis.

### 2.3. Performance Measurement

Body weight and backfat thickness of sows were measured on days 0 and 109 of pregnancy and at weaning (day 21 of lactation). Backfat thickness at 65 mm on each side of the dorsal midline at the last rib level (P2) was measured using ultrasound (Renco Lean-Meatier; Renco Corporation, Minneapolis, MN, USA). Within 12 h after farrowing, the number and the weight of live and dead piglets born per litter were recorded. Crossfostering was performed within 24 h after farrowing for each dietary treatment. Litters were crossfostered to adjust litter size to about 11.68 ± 0.89 piglets per sow, with no differences in the number of piglets among CON, RS and FSF after crossfostering. At weaning, the number of weaned piglets and weaning-to-estrus interval were recorded for each sow and litter. Piglets were weighed within 24 h of birth (day 0) and on days 7, 14 and 21 of lactation. The daily feed intake of sows during lactation was measured each morning by weighing daily feed refusals.

### 2.4. In Vitro Fermentation Analysis

Three gestation diets were incubated in duplicate with the in vitro two-stage enzyme incubation and dialysis procedures. The samples were subjected to in vitro chemical and enzymatic digestion using porcine pepsin and pancreatic enzymes as previously reported [22]. The in vitro fermentation followed the procedure as previously described for ruminants [23] with adaptation to the pig [24]. 

Briefly, fecal samples were opened inside an anaerobic chamber, and 20 g feces from each of the three pregnant sows (Landrace) from the herd of Swine Centre, Hunan Academy of Agricultural Sciences (Changsha, China) in a group were mixed together in a big container. Fecal slurry (10%) was prepared inside the anaerobic chamber by mixing 100 g of the pooled feces with 900 mL of buffer solution composed of macro and micro minerals, carbonate buffer, and reducing solution, stirred continuously to homogenize, and filtered through a double-layered cheese cloth [25]. The bottles were sealed with butyl rubber stoppers and aluminum caps to contain the gas pressure produced from fermentation, and incubated in a 39 °C incubator inside the anaerobic chamber for 72 h. In vitro fermentation of enzymatically hydrolyzed residues was then carried out as previously reported [24].

Kinetics of fermentation was assessed by measuring cumulative gas production over time and gas accumulation was modeled according to the method previously described [22]. After fermentation, the fermented broth was used for determination of short-chain fatty acids (SCFAs) using gas chromatography (Agilent 6890 series GC system; Agilent Technologies, Santa Clara, CA, USA) fitted with a flame ionization detector and a fused-silica capillary column [26].

### 2.5. Chemical Analysis and Calculation

The same 8 sows per group were used to analyze oxidative stress parameters on day 109 of gestation. Serum samples were used to measure the levels of protein carbonyl, 8-hydroxy-deoxyguanosine (8-OHdG), ferric reducing-antioxidant power (FRAP), and malondialdehyde (MDA) with specific assay kits (Nanjing Jiancheng Bioengineering Institute, Nanjing, China). The major marker of oxidative damage to nucleic acids, 8-OHdG, was measured by using an anti-8-OHdG monoclonal antibody in an enzyme-linked immunosorbent assay (ELISA) kit and expressed as ng 8-OHdG per milliliter of serum [27]. The activity of total antioxidant capacity (T-AOC) was measured at 520 nm by the method of FRAP assay. One unit of T-AOC was defined as the amount that increased the absorbance by 0.01 at 37 °C in 1 min [28]. MDA, as a product of lipid peroxidation, was measured by reaction with thiobarbituric acid at 95 °C and expressed as nmol MDA per milliliter of plasma [29]. The content of protein carbonyl was determined by derivatization using dinitrophenylhydrazine as reported previously [30].

The same 8 sows per group were used to analyze GLP-1, PYY, and cortisol on day 70 of gestation. Before analysis of GLP-1 and PYY, serum was thawed, and an additional protease inhibitor cocktail was added according to published method [20]. The concentrations of GLP-1 and PYY in serum were analyzed using an ultrasensitive pig GLP-1 and PYY enzyme-linked immunosorbent assay (ELISA) kit (CUSABIO, Wuhan, Hubei, China) according to the operation instructions. Cortisol concentrations in plasma and saliva were determined using a commercial ELISA kit (CUSABIO, Wuhan, Hubei, China) according to the manufacturer’s instructions.

### 2.6. Behavior Determination

The behaviors of twenty-four sows (eight replicates per dietary treatment) were observed at 2 h after feeding by time-lapse video recording for 1 h in the daily observation period on the morning of days 71, 72 and 73 of gestation. Postures evaluated included the duration of time spent in lying, sitting, or standing. Activities recorded included drinking, sham-chewing, sniffing, licking and position change. The number of water drinking bouts and body position change were recorded [8]. Sniffing, licking, and sham chewing were combined to form one category of stereotypic behaviors. The behavior time interval should be more than 8 s, and a behavior, if occurring again within 8 s and lasting 8 s or more, is determined to last 15 s. Based on this, the ratio of the initiation time of a single action to the total time was counted. Each sow was observed for a total of 60 min (5 min × 12 sessions) in each observation period and the ethogram is provided in Appendix A.

### 2.7. Statistical Analysis

Gas production parameters and fermentation metabolite production were analyzed using the MIXED procedure of statistical analysis system (SAS Institute Inc., Cary, NC, USA) according to published methods [31].

The number of sows during the experimental period is included in Appendix A. Individual sow was considered as the experimental unit in all statistical analyses. After the test data were sorted by Excel software, one-way analysis of variance was performed by SAS program. The growth performance of the piglets was analyzed by the mixed model variance component analysis. Before analysis, the data were tested for normality and homoscedasticity using the Kolmogorov-Smirnov and Levene tests (with the significance level set at 5%). Multiple comparisons were performed with Duncan multiple range test. The piglet stillbirth rate was analyzed using the Chi-square test. Differences between mean values were considered statistically significant at *p* < 0.05, with a trend toward significance at 0.05 < *p* < 0.10.

## 3. Results

### 3.1. Physical Properties of Gestation Diets and In Vitro Fermentability of Gestation Diets

As shown in Table 1, swelling capacity was higher in the RS diet than in the CON or FSF diet. In Table 2, the three dietary treatments were shown to have no significant differences (*p* > 0.05) in their effect on vitro fermentability of gestation diets. Meanwhile, the 48 h cumulative gas production and the final asymptotic gas volume showed an increased trend (*p* = 0.07, *p* = 0.09, respectively) in the RS group versus the CON or FSF group. Furthermore, the three groups exhibited no difference (*p* > 0.05) in their effect on concentrations of acetic acid, propionic acid, butyric acid, iso-butyric acid, valeric acid, iso-valeric acid, and total SCFA.

### 3.2. Sow Performance

In Table 3, the three dietary treatments were shown to have no obvious difference in body weight and backfat thickness of sows at mating, on day 109 of gestation, at farrowing and at weaning. The inclusion of fiber in the gestation diet of sows also had no effect on body weight or backfat thickness gain during gestation or their loss during lactation as well as weaning-to-estrus interval of sows (*p* > 0.05) (Table 3). Average daily feed intake throughout lactation exhibited no significant difference (*p* = 0.98) between the control group and the treatment groups (Table 3).

### 3.3. Piglet Performance

Table 4 shows the piglet performance in the three dietary treatments. The three groups had no significant differences (*p* > 0.05) (Table 4) in the numbers of total piglets born, born alive, after crossfostering, and at weaning, in contrast to a reduction (*p* < 0.05) in the number of stillbirths for the RS group versus the CON or FSF group, with a significantly lower (*p* < 0.01) stillbirth rate being observed in the former group (Figure 1). Meanwhile, the inclusion of resistant starch or fermented soybean fiber in the gestation diet of sows had no effect (*p* > 0.05) on litter weight, average piglet weight, and ADG of suckling piglets throughout lactation, with no significant differences (*p* > 0.05) being observed in each of the three indicators throughout this stage (Table 4).

### 3.4. Behavior Determination

In Table 5, sows fed RS diet were shown to spend less time on standing behavior (*p* < 0.05) than the CON or FSF group, while there were no differences (*p* > 0.05) among the three treatments in the sows’ physical activity (sitting, lying, kneeling and position change), drinking, sham-chewing, sniffing and licking behavior. However, there were huge, although not significant (*p* > 0.05), differences in the numerical values between the control diet and the RS diet in the sows’ behaviors of sham-chewing, sniffing and licking, and position change. Compared with the CON group, the RS group decreased the sham-chewing behavior by 69%, the sniffing and licking behavior by 65% and the position change behavior by twice (Table 5).

### 3.5. PYY and GLP-1 Concentration of Serum

As shown in Figure 2, GLP-1 and PYY concentrations of 2 h postprandial serum of sows on day 70 of gestation were significantly (*p* < 0.05) increased in the RS group versus the CON or FSF group. Meanwhile, sows’ pre-prandial serum GLP-1 concentrations exhibited an obvious increase (*p* < 0.05) in the RS and FSF group, in contrast to no difference (*p* > 0.05) observed in their pre-prandial serum PYY concentration among the three dietary treatments.

### 3.6. Cortisol Concentration in Plasma and Saliva

The three diet groups showed no significant differences (*p* > 0.05) in plasma and saliva concentration of sows’ pre-prandial cortisol on day 70 of gestation (Figure 3). However, compared with the CON group, the RS group showed a decreased tendency (*p* = 0.07) in the sows’ plasma cortisol concentration on day 70 of gestation.

### 3.7. Oxidative and Antioxidative Indicators

Figure 4 shows the oxidative and antioxidative indicators measured in sows’ serum. The RS and FSF groups exhibited an increase (*p* < 0.05) in FRAP concentration, in contrast to no obvious difference (*p* > 0.05) among the three groups in serum MDA concentration and 8-OHdG concentration on day 109 of gestation. Meanwhile, the concentration of serum protein carbonyl was higher (*p* < 0.05) in the CON group than in the other two groups on day 109 of gestation.

## 4. Discussion

The present experiment aimed to quantify the effects of two available unconventional dietary fiber resources on sows’ reproductive performance through an in vitro-in vivo method. The inclusion of 5% RS in the diets of pregnant sows was observed to enhance the postprandial satiety, alleviate the stress status, reduce the abnormal behaviors and lower the stillbirth rate of sows.

Several studies have reported that restricted feeding induced abnormal behaviors in individually housed sows [5]. Abnormal behaviors, including stereotypic behaviors and abnormal physical activities, may reflect animal satiety [32]. In the current study, while stereotypic behaviors were not affected by diets, the standing behavior was lower, which means more rest time and lower feeding motivation in sows fed RS diet, indicative of a greater degree of postprandial satiety. This finding corresponds to a previous study of decreased standing [33]. Previous studies have shown that dietary fiber can affect the satiety of animals after feeding by delaying gastric emptying and stimulating the stomach wall via its swelling capacity [34], and the great swelling capacity of fiber could increase chewing activity and saliva production in the mouth [35], which can enhance postprandial satiety via the central nervous system [36]. In the present study, the diet including 5% RS showed a higher swelling capacity (2.28 mL/g), which means that the total dietary volume for the pregnant sows fed 2.4 kg/d of feed during gestation could reach 5.47 L. A possible explanation for the reduced standing behavior is the increase of postprandial satiety due to the larger dietary volume provided by RS in the feed.

Furthermore, feeding sows with 5% RS increased the serum concentrations of 2 h postprandial GLP-1 and PYY of pregnant sows, which may also result in enhanced postprandial satiety. Previous studies have shown that GLP-1 and PYY are important regulators of satiety and stereotypes [21,37], and they can reduce gut motility, delay gastric emptying, slow transit time to enhance digestion and nutrient absorption, and thus reduce appetite, implying that GLP-1 may influence eating at a later meal and hunger in the inter-meal period [38,39]. Long-term inclusion of RS in the diets of rats constantly up-regulated plasma GLP-1 concentration and large intestinal GLP-1 gene expression within 24 h [40]. Interestingly, GLP-1 level was also significantly increased before meals in the RS and FSF groups in our results. During the early postprandial period, GLP-1 production was predominantly sustained via glucose, and the increase before meals might be associated with the fermentation of RS in the lower gut [41]. A previous study also mentioned that microbiota in the distal part of the gastrointestinal tract can ferment dietary fiber to produce SCFAs, especially propionic acid, which could activate the gene expression of free fatty acid receptors in L-cells and promote the generation of GLP-1 and PYY in rodents and humans [42]. Nevertheless, in the present study, the three diets exhibited no significant effect on fermentability, except for the higher gas production values, probably due to the low dosage of RS. Another possible explanation is the limitations of in vitro enzymatic hydrolysis, which cannot completely mimic the digestive process in the body [43,44]. In a previous study on gestating sows, feeding a RS diet was shown to induce satiety by supplying a constant concentration of glucose for a longer time and increasing the concentration of nonestesterified fatty acid in blood [17]. In the present study, we did not measure the levels of these metabolites in the blood, and therefore we are unable to tell whether glucose and nonestesterified fatty acid would increase as noted by other researchers. Overall, the RS group had the highest levels of satiety hormones before or after a meal, which indicates a reduction in the hunger of sows.

Moreover, including 5% RS in the gestation diet showed a decrease in stillbirth relative to the CON or FSF group, with the numerical regularity of satiety and abnormal behaviors being consistent with that of stillbirth rate in the three treatment groups. These results agreed with previous studies reporting that frequent chewing behavior during pregnancy reduced the total litter size [2].

Our results also showed that the RS group alleviated stress status. Previous studies have indicated that sows suffer from severe oxidative stress during pregnancy [45] and alleviating oxidative stress could definitely benefit sow’s health status [46]. Accumulation of reactive oxygen species (ROS) may cause damage to macromolecules, including DNA, bio-membrane lipids, and proteins, corresponding to impairment to tissues. Systemic oxidative status can be determined by the oxidant and antioxidant parameters in the serum of sows, including T-AOC, 8-OhdG, protein carbonyl and MDA. T-AOC is associated with the elimination of free radicals and ROS, blocking peroxidation and thus preventing lipid peroxidation and removing catalytic metal ions [47]. Oxidative products of protein injury are generally reflected by the content of protein carbonyl [48]. The number of stillbirths has been reported to be positively correlated with a higher stress status on late pregnancy of sows [4,11], which is well supported by the higher level of serum T-AOC, the lower level of serum protein carbonyl, and a decline in stillbirth number in the RS group in the present study. Additionally, the secretion of stress hormone cortisol can be enhanced by abnormal behavior in pigs [49]. In the present study, the RS group showed a decreased tendency versus the CON group in the sows’ plasma cortisol concentration on day 70 of gestation. It also showed a decrease in stillbirth relative to the CON or FSF group, with the numerical regularity of stress being consistent with that of stillbirth rate in the three treatment groups, which was in line with an obvious positive correlation between stillbirth and stress status reported in a previous study [50]. Overall, these results indicate that feeding sows with 5% RS decreased the stress levels of pregnant sows, which may contribute to reducing stillbirth.

In the present study, the gestation diets exhibited no effects on sows’ weight and backfat during gestation and lactation, piglet BW, or weaning-to-estrus interval. Several factors may have converged to prevent RS from influencing reproduction performance of sows. Likewise, a previous study found no improvement in sow or litter performance when soybean hulls were offered to sows beginning 2 day postmating to day 109 of gestation [51]. Similar levels of lowly digestible fibers in the three diets of our study could be one of the reasons for fewer differences observed in the reproductive performance. Our research results are not in line with a previous study showing that feeding sows with a high-swelling diet containing konjac flour fiber during gestation increased the subsequent lactation feed intake of sows and tended to improve the performance of piglets [10], but the response is variable [52]. Timing of fiber addition to sow diets might be important. Several researchers began to feed fiber diets at breeding and continued throughout gestation over multiple parities, and reported increases in litter size after 2 or 3 parities [53,54]. Finally, the average parity of the sows reached the fifth parity in our study, and the excessive age of the sows may also mask the beneficial effects of RS [8]. However, few reports have shown that the high swelling diet with the inclusion of RS during one reproduction cycle could reduce the stillbirth rate. Our findings point to the importance that inclusion of 5% RS during gestation can have a positive effect on sows.

## 5. Conclusions

The present research indicated that inclusion of 5% resistant starch with greater swelling capacity in the gestation diet was shown to enhancing the postprandial satiety, alleviating the stress status, reducing the abnormal behaviors and thus lowering the stillbirth rate of sows.

## Figures and Tables

**Figure 1 animals-10-00141-f001:**
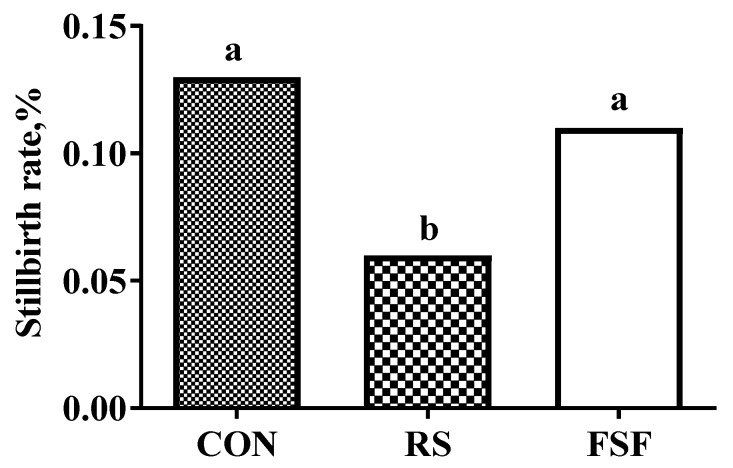
Effect of inclusion of resistant starch (RS) or fermented soybean fiber (FSF) in gestation diet on stillbirth rate. CON = control diet group; RS = 5% resistant starch diet group; FSF = 5% fermented soybean fiber diet group. Stillbirth rate was analyzed using the Chi-square. test. ^a,b^ Means with different superscripts within a row differ significantly (*p* < 0.05).

**Figure 2 animals-10-00141-f002:**
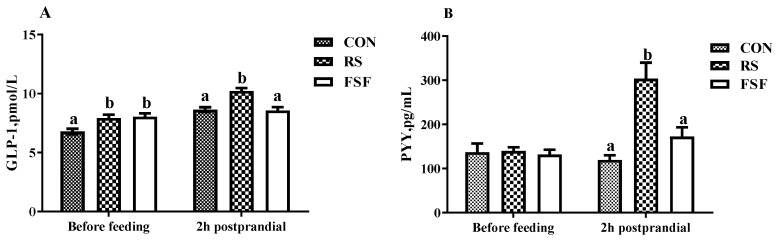
Effect of inclusion of resistant starch (RS) or fermented soybean fiber (FSF) in gestation diet on serum concentrations of sows’ GLP-1 (**A**) and PYY (**B**) on day 70 of gestation. CON = control diet group; RS = 5% resistant starch diet group; FSF = 5% fermented soybean fiber diet group. Values are means ± SEM (*n* = 7–8). ^a,b^ Means with different superscript letters within a row differ significantly (*p* < 0.05).

**Figure 3 animals-10-00141-f003:**
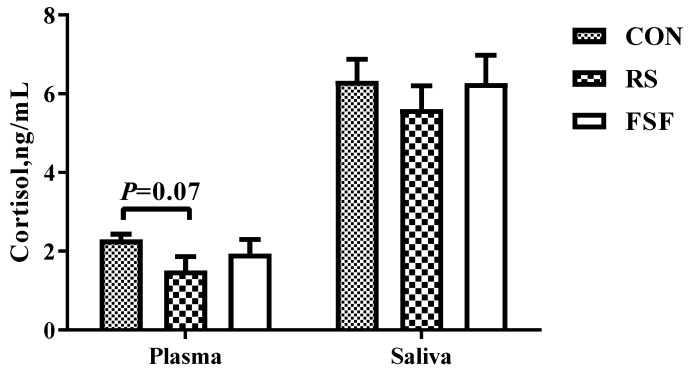
Effect of inclusion of resistant starch (RS) or fermented soybean fiber (FSF) in gestation diet on sows’ plasma and saliva cortisol concentration on day 70 of gestation. CON = control diet group; RS = 5% resistant starch diet group; FSF = 5% fermented soybean fiber diet group. Values are means ± SEM (*n* = 7–8).

**Figure 4 animals-10-00141-f004:**
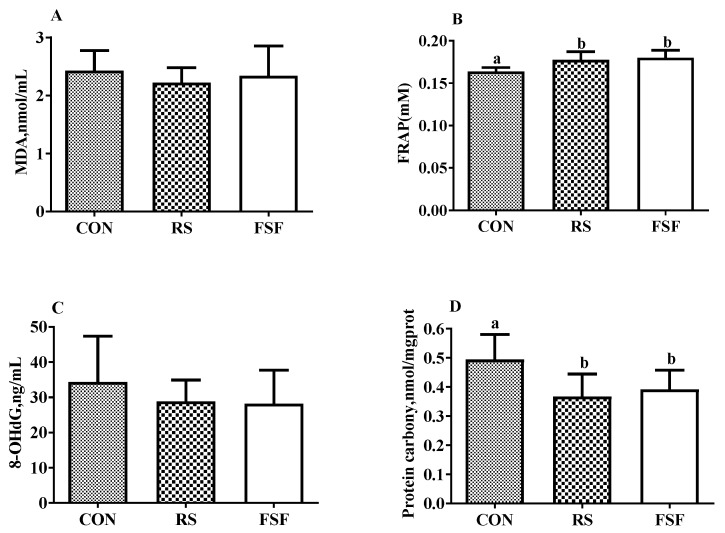
Dietary effects on sows’ serum levels of MDA (**A**), FRAP (**B**), 8-OHdG (**C**) and protein carbonyl (**D**) on day 109 of gestation. MDA = Malondialdehyde; 8-OHdG = 8-hydroxy-deoxyguanosine; FRAP = Ferric ion reducing antioxidant power. CON = control diet group; RS = 5% resistant starch diet group; FSF = 5% fermented soybean fiber diet group. Values are means ± SEM (*n* = 8). ^a,b^ Means with different superscript letters within a row differ significantly (*p* < 0.05).

**Table 1 animals-10-00141-t001:** Ingredients and nutrient composition of experimental gestation diets and lactation diets (as-fed basis).

Item	Gestation Diet			Lactation Diet
	CON ^1^	RS ^1^	FSF ^1^	
Ingredient, %				
Corn	61.60	53.60	61.30	61.00
Soybean meal	11.00	12.00	9.20	19.50
Wheat bran	20.00	20.00	18.00	-
Soybean hull	3.00	5.00	2.00	5.00
Wheat middlings	-	-	-	1.50
Corn meal				0.35
Fish meal, 67%CP				1.00
Extruded soybean				5.50
Soybean oil				0.60
Glucose				1.25
Resistant starch ^2^	-	5.00	-	-
Fermented soybean fiber ^3^	-	-	5.00	-
Dicalcium phosphate	1.70	1.70	1.70	1.55
Limestone	1.00	1.00	1.00	0.90
Salt	0.50	0.50	0.50	0.50
Lysine sulfate (70%)	0.20	0.20	0.30	0.30
Threonine				0.10
Methionine				0.05
Choline chloride	0.10	0.10	0.10	0.10
Sodium bicarbonate	0.40	0.40	0.40	0.40
Mildewcide	0.10	0.10	0.10	-
Premix ^4^	0.40	0.40	0.40	0.40
Calculated compositon ^5^				
DE (Mcal/kg)	2.99	2.98	3.00	3.30
CP (%)	13.30	13.30	13.30	17.61
EE (%)	3.10	2.90	3.10	4.05
CF (%)	5.40	6.00	6.33	3.80
NDF (%)	17.00	17.80	17.90	12.26
Ca (%)	0.80	0.90	0.80	0.85
Total phosphorus (%)	0.70	0.70	0.70	0.64
Lys (%)	0.70	0.70	0.70	1.15
Met (%)	0.20	0.20	0.20	0.34
Thr (%)	0.50	0.50	0.50	0.78
Trp (%)	0.20	0.20	0.20	0.19
Analyzed composition				
CF (%)	5.34	5.69	6.30	
NDF (%)	19.00	19.15	19.50	
Physio-chemical properties				
Viscosity (mPa·s)	1.83	1.88	1.82	-
Water-blinding capacity (g/g)	0.09	0.09	0.10	-
Swelling capacity (mL/g)	1.63	2.28	1.89	-

^1^ CON = control diet group; RS = 5% resistant starch diet group; FSF = 5% fermented soybean fiber diet group; ^2^ Resistant starch contains 3.69 Mcal/kg gross energy, 0.39% crude protein (CP), 1.22% crude protein (CF), and 19.15% neutral detergent fibre (NDF); ^3^ Fermented soybean fiber contains 4.08 Mcal/kg gross energy, 28.11% crude protein (CP), 37.63% crude fiber (CF), and 76% neutral detergent fibre (NDF); ^4^ Provided per kg of diet: Fe 145 mg as ferrous sulfate; Zn 75 mg as zinc sulfate; Mn 50 mg (as MnO_2_); Cu 10.0 mg (as CuSO_4_·5H_2_O); Se 0.3 mg selenium selenite; I 0.25 mg as potassium iodide; Gu 0.1 mg; Vitamin A 7,500 IU; Vitamin D3 4,992 IU; vitamin E 215.2 mg; Vitamin C 200 mg; Niacin 50 mg; 22 mg riboflavin; 8.5 mg pyridoxine; Vitamin K 35.1 mg; Folic acid 4.5 mg; Ammonium 3.7 mg; ^5^ Calculated chemical concentrations using values for feed ingredients from (NRC, 2012) [19].

**Table 2 animals-10-00141-t002:** Gas production parameters and concentrations of short-chain fatty acids (SCFA) during the in vitro fermentation of enzymatically hydrolyzed residues of the three diets using fecal inocula from gestation sows.

Item	Diet			SEM	*p*-Value
	CON ^1^	RS ^1^	FSF ^1^		
Fermented Fraction					
V ^2^	187.17	197.60	169.99	5.60	0.07
V_F_ ^3^	210.41	222.77	197.96	5.08	0.09
(FRD_0_) ^4^ ×100	1.86	2.00	2.31	0.16	0.61
K ^5^	0.10	0.08	0.06	0.01	0.65
(T_1/2_) ^6^	20.75	20.52	20.93	0.24	0.86
Concentrations of SCFA					
Acetic acid, mmol/L	20.08	19.65	16.15	1.15	0.39
Propionic acid, mmol/L	9.17	8.57	6.67	0.64	0.30
Butyric acid, mmol/L	1.59	1.35	1.18	1.10	0.32
*Iso*-butyric acid, mmol/L	0.49	0.30	0.26	0.06	0.31
Valeric acid, mmol/L	1.50	1.09	0.97	0.13	0.22
*Iso*-valeric acid, mmol/L	0.83	0.46	0.40	0.12	0.33
Total SCFA ^7^, mmol/L	33.66	31.43	25.63	2.07	0.32

^1^ CON = control diet group; RS = 5% resistant starch diet group; FSF = 5% fermented soybean fiber diet group; *n* = 2 (Number of observations in fermentation); ^2^ V, 48 h cumulative gas production; ^3^ VF, The final asymptotic gas volume (mL/g); ^4^ FRD0, Initial fractional rate of degradation at *t*-value = 0 (h-1); ^5^ K, Fractional rate of gas production at a particular time point (h-1); ^6^ T1/2, Half-life to asymptote (h); ^7^ Total SCFA = the sum of acetic, propionic, butyric, iso-butyric, valeric and iso-valeric acid.

**Table 3 animals-10-00141-t003:** Effect of inclusion of resistant starch (RS) or fermented soybean fiber (FSF) in gestation diet on body weight, backfat thickness, weaning to estrus interval and feed intake during lactation of sows.

Item	Diet			SEM	*p*-Value
	CON ^1^	RS ^1^	FSF ^1^		
No. of sows	22	21	24		
Average sow parity	5.1	4.9	4.9		
BW of sows, kg					
Mating	229.2	228.9	229.9	2.89	0.99
D109 of gestation ^2^	285.5	279.1	282.8	2.84	0.62
Gain during gestation	57.6	51.0	51.2	1.64	0.19
Parturition ^2^	258.5	255.6	257.1	2.82	0.96
Weaning	243.9	243.8	240.6	2.97	0.83
Loss during lactation ^2^	16.4	12.4	18.3	1.64	0.45
Sow backfat thickness, mm					
Mating	17.5	17.6	17.1	0.52	0.91
D109 of gestation	17.6	16.9	17.4	0.55	0.83
Weaning ^2^	16.2	15.6	15.2	0.42	0.52
Weaning to estrus interval ^2^, d	4.2	4.3	4.8	0.13	0.21
Average daily feed intake ^3^, kg					
1st week of lactation	5.3	5.6	5.5	0.14	0.89
2nd week of lactation ^2^	7.2	7.3	7.3	0.11	0.92
3rd week of lactation	7.8	7.3	7.5	0.12	0.27
Mean of 1st week to 3rd week	6.8	6.7	6.7	0.09	0.98

^1^ CON = control diet group; RS = 5% resistant starch diet group; FSF = 5% fermented soybean fiber diet group; ^2^ Results were analyzed using the Kruskal-Wallis test; ^3^ Number of sows in CON, RS and FSF is 19, 21 and 22, respectively.

**Table 4 animals-10-00141-t004:** Effect of inclusion of resistant starch (RS) or fermented soybean fiber (FSF) in gestation diet on litter performance.

Item	Diet			SEM	*p*-Value
	CON ^1^	RS ^1^	FSF ^1^		
No. of sows	19	21	22		
Average sow parity	4.5	4.8	5.0		
No. of pigs per litter					
Total piglets born ^2^	15.6	14.0	15.0	0.38	0.24
Piglets born alive ^2,3^	13.3	13.0	13.0	0.33	0.77
Stillbirth ^2,3^	2.0 ^a^	0.9 ^b^	1.8 ^ab^	0.20	0.04
After cross-foster ^3^	12.1	11.6	11.5	0.11	0.12
Pigs weaned ^3^	10.6	10.0	10.3	0.18	0.20
Piglet mean BW, kg					
At Birth ^2,3^	1.4	1.4	1.4	0.03	0.99
After cross-foster ^3^	1.6	1.6	1.6	0.03	0.43
On day 7	3.1	3.2	3.1	0.06	0.59
On day 14	5.1	5.1	5.1	0.05	0.67
On day 21	7.0	7.1	7.0	0.07	0.73
Average daily gain, g/d					
Day 1 to 7	209.7	220.3	207.0	7.90	0.45
Day 7 to 14	277.9	283.9	274.4	7.54	0.67
Day 14 to 21	268.0	276.4	261.8	11.00	0.65
Day 1 to 21	253.6	265.8	249.5	6.80	0.22

^1^ CON = control diet group; RS = 5% resistant starch diet group; FSF = 5% fermented soybean fiber diet group; ^2^ Number of sows in CON, RS and FSF is 22, 21 and 24, respectively; ^3^ Results were analyzed using the Kruskal-Wallis test; ^a,b^ Means with different superscript letters within a row differ significantly (*p* < 0.05).

**Table 5 animals-10-00141-t005:** Effect of inclusion of resistant starch (RS) or fermented soybean fiber (FSF) in gestation diet on behaviors of pregnant sows.

Item	Diet			SEM	*p*-Value
	CON ^1^	RS ^1^	FSF ^1^		
No. of sows	8	8	8		
Lying	91.88	96.46	91.20	1.45	0.13
Standing	7.66 ^a^	1.77 ^b^	7.24 ^a^	1.31	0.05
Sitting	0.47	1.77	1.62	1.38	0.74
Drinking, times	2.00	1.25	1.63	0.31	0.45
Sham-chewing	13.07	4.11	14.32	3.41	0.24
Sniffing, licking	1.93	0.68	0.63	0.27	0.19
Position change ^2^, times	3.63	1.75	3.38	0.44	0.17

^1^ CON = control diet group; RS = 5% resistant starch diet group; FSF = 5% fermented soybean fiber diet group; ^2^ Results were analyzed using the Kruskal-Wallis test; ^a,b^ Means with different superscript letters within a row differ significantly (*p* < 0.05).

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
