# Peer review of "Effects of Dietary Fiber Sources during Gestation on Stress Status, Abnormal Behaviors and Reproductive Performance of Sows"

_animals, 2020, doi:10.3390/ani10010141_

Round 1

Reviewer 1 Report

Reviewer #2 (Comments for the Author): 
The manuscript by Huang et al reports on interesting finding in relation to the dietary fiber to enhance the postprandial satiety, alleviate stress status and reduce stereotypic behaviors and thus stillbirth rate of sows. The data are of value for scientist and farmers alike, especially at this time, African swine fever is in an outbreak in China cause enormous economic loss in swine.The manuscript has merit, but there are certain areas that need to be improved in order to be considered for publication. 

How is the amount of dietary fiber supplementation determined? L94-96: Were allotments of feed weighed daily or were feed allowances determined volumetrically? If the latter, how often were feed drops calibrated. I expect there was a large difference in bulk density of diets. L99-100: Did piglets have access to sow feed? It will be better to have a good explanation for the significant of GLP-1 and PYY before and after meals? In the results section, the authors need to present their data by not only stating the significance but also discussing it to combine biological events. If not, it may be not easy to understand. I think this experiment may be benefit for pig production. To be better understood for readers, I strongly suggest the author to re-write abstract and discussion sections, because I can’t totally get the meaning to replace the diet with these two stuffs without so strong output.

Author Response

We would like to thank you for the constructive advice on how to improve the quality of the paper.The following is a point-to-point response to the reviewers’ comments.

Reviewer #1 (Comments for the Author):

The manuscript by Huang et al reports on interesting finding in relation to the dietary fiber to enhance the postprandial satiety, alleviate stress status and reduce stereotypic behaviors and thus stillbirth rate of sows. The data are of value for scientist and farmers alike, especially at this time, African swine fever is in an outbreak in China cause enormous economic loss in swine.The manuscript has merit, but there are certain areas that need to be improved in order to be considered for publication.

Response: Thank you for your positive comments and valuable suggestions on our manuscript. We have carefully considered the comments and revised the manuscript accordingly.

How is the amount of dietary fiber supplementation determined?

Response: Thank you for your comment. Fiber intake during gestation for sows was widely calculated as daily neutral detergent fiber (NDF). Reese (2008) evaluated the amount of fiber added to the corn and soybean meal-type diets by examining 24 reports published from 1975 to 2007, and then suggested that up to 380 grams of NDF/day may optimize litter size [1]. In addition, a previous study has shown that resistant starch diet (inclusion of 17% NDF) can improve the welfare of sows by reducing aggression and increasing satiety in limit-fed pregnant sows without affecting reproductive performance [2]. Meanwhile, feeding diets with high-RS content to gestating sows reduced the birth weight of piglets [3]. Thus, we speculated that supplementation of 5% RS (442 g/d NDF) to gestation diet may have a positive effect on sows and their offspring’s performance and health. To our best knowledge, no report has been published on the effect of dietary addition of fermented soybean fiber on sows, and the amount of RS and FS supplementation proposed in this was consistent with each other based on the aforementioned studies.

Reference:

Reese D, Prosch A, Travnicek D A, et al. Dietary fiber in sow gestation diets-an updated review[J]. 2008. Sapkota A, Marchant-Forde J N, Richert B T, et al. Including dietary fiber and resistant starch to increase satiety and reduce aggression in gestating sows[J]. Journal of animal science, 2016, 94(5): 2117-2127. Yan H, Lu H, Almeida V V, et al. Effects of dietary resistant starch content on metabolic status, milk composition, and microbial profiling in lactating sows and on offspring performance[J]. Journal of animal physiology and animal nutrition, 2017, 101(1): 190-200.

L94-96: Were allotments of feed weighed daily or were feed allowances determined volumetrically? If the latter, how often were feed drops calibrated. I expect there was a large difference in bulk density of diets.

Response: The amount of feed during gestation was weighed twice daily with plastic bags and correspondingly put into a barrel before feeding. Thank you.

L99-100: Did piglets have access to sow feed?

Response: No. sow feed was separately contained in the trough, and it was not accessible to the piglets during entire lactation. Please see L103-104. Thank you.

It will be better to have a good explanation for the significant of GLP-1 and PYY before and after meals?

Response: Thank you for your comment. We believe that GLP-1 and PYY both regulate satiety before and after meals. However, the maintenance factors of GLP-1 and PYY levels are different in the early and longer periods of eating. During the early postprandial period, GLP-1 production was predominantly sustained via glucose, while GLP-1 and PYY reduced gut motility, delayed gastric emptying and slowed transit time to enhance digestion and nutrient absorption and thus reduced appetite, implying that GLP-1 may influence eating at a later meal and hunger in the inter-meal period [1, 2]. Interestingly, GLP-1 level was also significantly increased before meals in the RS and FSF groups in our results which might be associated with the fermentation of RS in the lower gut [3]. A previous study has shown that long-term feeding of RS rats continuously un-regulated plasma GLP-1 and PYY concentration and large intestinal GLP-1 and PYY gene expression within 24 hours [4]. A previous study also mentioned that microbiota in the distal part of the gastrointestinal tract can ferment dietary fiber to produce SCFAs, especially propionic acid, which could activate the gene expression of free fatty acid receptors in L-cells and promote the generation of GLP-1 and PYY in rodents and humans [5]. Nevertheless, in the present study, the three diets exhibited no significant effect on fermentability, except for the higher gas production values, probably due to the low dosage of RS. Another possible explanation is the limitations of in vitro enzymatic hydrolysis, which cannot completely mimic the digestive process in the body [6, 7]. In addition, feeding a RS diet was shown to induce satiety by supplying a constant concentration of glucose for a longer time and increasing the concentration of nonestesterified fatty acid in blood in a previous study on gestating sows [8]. We did not measure the levels of these metabolites in the blood, and therefore we are unable to know if glucose and nonestesterified fatty acid in our study would increase as noted by other researchers. Overall, the RS group had the highest levels of satiety hormones before or after a meal, which indicates a reduction in hunger of sows. Please refer to L338-361 in discussion. Thank you.

Reference:

Blundell, J.E.; Naslund, E. Glucagon-like peptide-1, satiety and appetite control. Br J Nutr. 1999, 81, 259-60. Sleeth, M.L.; Thompson, E.L.; Ford, H.E.; Zac-Varghese, S.E.; Frost, G. Free fatty acid receptor 2 and nutrient sensing: a proposed role for fibre, fermentable carbohydrates and short-chain fatty acids in appetite regulation. Nutrition research reviews. 2010, 23, 135-45. Zhou, J.; Martin, R.J.; Tulley, R.T.; Raggio, A.M.; McCutcheon, K.L.; Shen, L.; Danna, S.C.; Tripathy, S.; Hegsted, M.; Keenan, M.J. Dietary resistant starch upregulates total GLP-1 and PYY in a sustained day-long manner through fermentation in rodents. American Journal of Physiology-Endocrinology and Metabolism. 2008, 295, E1160-E6. Zhou, J.; Keenan, M.J.; Raggio, A.M.; Tripathy, S.; Shen, L.; McCutcheon, K.L.; Hegsted, M.; Tulley, R.T.; Martin, R.J. Feeding resistant starch maintains elevated plasma levels of GLP-1 and PYY throughout the day and is associated with decreased body fat in rats. Federation of American Societies for Experimental Biology; 2007. Byrne, C.; Chambers, E.; Morrison, D.; Frost, G. The role of short chain fatty acids in appetite regulation and energy homeostasis. International journal of obesity. 2015, 39, 1331. Coles, L.; Moughan, P.; Darragh, A. In vitro digestion and fermentation methods, including gas production techniques, as applied to nutritive evaluation of foods in the hindgut of humans and other simple-stomached animals. Animal Feed Science and Technology. 2005, 123, 421-44. Bindelle, J.; Buldgen, A.; Lambotte, D.; Wavreille, J.; Leterme, P. Effect of pig faecal donor and of pig diet composition on in vitro fermentation of sugar beet pulp. Animal feed science and technology. 2007, 132, 212-26. Sapkota, A.; Marchant-Forde, J.N.; Richert, B.T.; Lay, D.C. Including dietary fiber and resistant starch to increase satiety and reduce aggression in gestating sows. Journal of Animal Science. 94, 2117.

In the results section, the authors need to present their data by not only stating the significance but also discussing it to combine biological events. If not, it may be not easy to understand.

Response: Thank you for your comment. We have rewritten the discussion section. In the discussion section, we present the results and discuss them to combine biological events. Please see the discussion section.

I think this experiment may be benefit for pig production. To be better understood for readers, I strongly suggest the author to re-write abstract and discussion sections, because I can’t totally get the meaning to replace the diet with these two stuffs without so strong output.

Response: Thank you for your comment. We have rewritten them accordingly. Few reports have shown that the high swelling capacity diet with the inclusion of resistant starch could reduce stillbirth rate. Our findings point to the importance that inclusion of 5% resistant starch with greater swelling capacity in the gestation diet was beneficial to enhancing postprandial satiety, alleviating stress status, reducing abnormal behaviors and thus lowering the stillbirth rate of sows. Please see abstract and discussion sections.

Reviewer 2 Report

The paper entitled “Effects of dietary fiber sources during gestation of sows on their stress status, stereotypic behaviors, and reproductive performance” was meant to provide insight on the effect different fiber sources on sow behavior and reproductive performance. I regret that the paper cannot be accepted for publication due to open blanket statements that are not substantiated, language issues, the introduction leading to study objective is not focused, lack of clarity in sample preparation, inconsistent data, contradictions, as well as comments, questions and concerns stated below.

Lines 2 to 4: The title of the paper “Effects of dietary fiber sources during gestation of sows on their stress status, stereotypic behaviors, and reproductive performance” is inappropriate. Suggested title could be “Effects of dietary fiber sources during gestation on stress status, stereotypic behaviors, and reproductive performance of sows”.

Line 45: Why do authors think that stress is considered as stereotypic behavior?

Line 47: Be specific about abnormal behavior that reduced total litter size.

Lines 50 to 51: Fiber is not fed to pregnant sows to save feed sources. Please, review the benefits of fiber to pregnant sows.

Lines 80 to 82: How do authors determine sow reproductive performance through chemical evaluation?

Line 94: What is digestive energy?

Line 107: Be specific about the oxidative stress parameters to help the readers.

Line 126: Piglets were weighed at weaning but weaning data was not provided in table 4.

Lines 161 to 169: The blood samples for GLP-1 and peptide YY how were they treated or prepared before the ELISA procedure?

Lines 194 to 195: What is the p-value associated with that statement?

Lines 196 to 198: Where is the data to support that result?

Line 210: Correct compound name as CuSO4·5H2O but not CuSO4·5H2O

Line 238: Information about sow numbers contradicts what is in table 3.

Line 253: Information about sow numbers contradicts what is in table 4.

Line 250. The sum of piglets born alive and stillborn do not add up to the total piglets born, table 4.

Line 250: Piglets mean BW and litter weight provide the same information and therefore repetitive.

Lines 107 and 148: Blood samples were collected on day 109 of gestation but this contradicts the information on lines 278 and 284.

Lines 310 to 313; Lines 80 to 82: Depending on which section of the manuscript you read, different objectives are stated.

Lines 325 to 326: “Furthermore, including 5% resistant starch fiber in the gestation diet increased the serum concentration of GLP-1”. The baseline values were different for GLP-1. Will the result above be the same if authors had accounted for the baseline values at 2 hour postprandial? Also only one blood sample was collected from 24 fasted sows according to lines 105 to 110. How did authors obtain blood samples before and after feeding?

In general, the manuscript lacks focus and clarity in the methodology. Additionally, it contains inconsistent data and contradictions.

Author Response

We would like to thank you for the constructive advice on how to improve the quality of the paper. The following is a point-to-point response to the reviewers’ comments.

Reviewer #2 (Comments for the Author):

Lines 2 to 4: The title of the paper “Effects of dietary fiber sources during gestation of sows on their stress status, stereotypic behaviors, and reproductive performance” is inappropriate. Suggested title could be “Effects of dietary fiber sources during gestation on stress status, stereotypic behaviors, and reproductive performance of sows”.

Response: Thank you for your comment. We have rewritten the title. Please see L2-4.

Line 45: Why do authors think that stress is considered as stereotypic behavior?

Response: We are so sorry for the confusing statement. We have revised them accordingly. Excessive restriction of feeding makes the sows less satisfied during pregnancy, leading to the frequent occurrence of stereotypes, which will be accompanied by the increases of stress and physical harm, and in turn a negative impact on their reproductive performance [1]. Please see L48-51.

Reference:

Robert, S.; Bergeron, R.; Farmer, C.; Meunier-Salaün, M.C. Does the  number of daily meals affect feeding motivation and behaviour of gilts fed high-fibre diets? Applied Animal Behaviour Science. 2002, 76, 105-17.

Line 47: Be specific about abnormal behavior that reduced total litter size.

Response: Many thanks to you for your reminder. We are sorry for not presenting the information clearly. Previous studies have shown that sows with a high relative frequency of vacuum chewing during gestation produced fewer total pigs born and a decrease of embryo survival than those with no vacuum chewing [1-3]. Please see L51-53 in introduction.

Reference:

Robert, S.; Bergeron, R.; Farmer, C.; Meunier-Salaün, M.C. Does the number of daily meals affect feeding motivation and behaviour of gilts fed high-fibre diets? Applied Animal Behaviour Science. 2002, 76, 105-17. Renteria-Flores, J.; Johnston, L.; Shurson, G.C.; Moser, R.; Webel, S. Effect of soluble and insoluble dietary fiber on embryo survival and sow performance. J Anim Sci. 2008, 86, 2576-84. Sekiguchi T, Koketsu Y. Behavior and reproductive performance by stalled breeding females on a commercial swine farm[J]. Journal of animal science, 2004, 82(5): 1482-1487.

Lines 50 to 51: Fiber is not fed to pregnant sows to save feed sources. Please, review the benefits of fiber to pregnant sows.

Response: We have rewritten the part based on your suggestions. Inclusion of fiber in gestation diets is not only a method for promoting nutritional satiety and reducing apparent feeding motivation in sows without providing excess energy, but also has the potential to increase feed intake during lactation of sows. Please see L53-56. Thank you.

Lines 80 to 82: How do authors determine sow reproductive performance through chemical evaluation?

Response: Thank you for your comment. We revised it accordingly. Please see L85-87.

Line 94: What is digestive energy?

Response: We apologize for this mistake. We have revised it accordingly. Please see L99.

Line 107: Be specific about the oxidative stress parameters to help the readers.

Response: Thank you for your comment. We have rewritten the part based on your suggestions. We have specified about oxidative stress parameters in the chemical analysis and calculation of materials and methods, and discussion section. The same 8 sows per group were used to analyze oxidative stress parameters on day 109 of gestation. Serum samples were used to measure the levels of protein carbonyl, 8-hydroxy-deoxyguanosine (8-OHdG), ferric reducing-antioxidant power (FRAP), and malondialdehyde (MDA) with specific assay kits (Nanjing Jiancheng Bioengineering Institute, Nanjing, China). The major marker of oxidative damage to nucleic acids, 8-OHdG, was measured by using an anti-8-OHdG monoclonal antibody in an enzyme-linked immunosorbent assay (ELISA) kit and expressed as ng 8-OHdG per millilitre of serum [1]. The activity of T-AOC was measured at 520 nm by the method of FRAP assay. One unit of T-AOC was defined as the amount for increasing the absorbance by 0.01 at 37 °C in 1 min [2]. MDA, as a product of lipid peroxidation, was measured by reaction with thiobarbituric acid at 95 °C, and expressed as nmol MDA per milliliter of plasma [3]. The content of protein carbonyl was determined by derivatization using dinitrophenylhydrazine as reported previously [4].

    Alleviating oxidative stress could definitely benefit sow’s health status [5]. Accumulation of reactive oxygen species (ROS) may cause damage to macromolecules, including DNA, bio-membrane lipids, and proteins, corresponding with impairment to tissues. Systemic oxidative status was determined by measuring the oxidant and antioxidant parameters in the serum of sow, including T-AOC, 8-OhdG, protein carbonyl and MDA. T-AOC is associated with the elimination of free radicals and ROS, blocking peroxidation and thus preventing lipid peroxidation and removing catalytic metal ions [6]. Oxidative products of protein injury are generally reflected by the content of protein carbonyl [7]. Please see L158-169 in meterials and methods and L368-374 in discussion section.

Reference:

Van, P.Y.; Hamilton, G.J.; Kremenevskiy, I.V.; Sambasivan, C.; Spoerke, N.J.; Differding, J.A.; Watters, J.M.; Schreiber, M.A. Lyophilized plasma reconstituted with ascorbic acid suppresses inflammation and oxidative DNA damage. Journal of Trauma and Acute Care Surgery. 2011, 71, 20-5. Benzie, I.F.; Strain, J.J. The ferric reducing ability of plasma (FRAP) as a measure of “antioxidant power”: the FRAP assay. Anal Biochem. 1996, 239, 70-6. Grotto, D.; Santa Maria, L.; Boeira, S.; Valentini, J.; Charão, M.; Moro, A.; Nascimento, P.; Pomblum, V.; Garcia, S. Rapid quantification of malondialdehyde in plasma by high performance liquid chromatography–visible detection. J Pharm Biomed Anal. 2007, 43, 619-24. Ganhão, R.; Morcuende, D.; Estévez, M. Protein oxidation in emulsified cooked burger patties with added fruit extracts: influence on colour and texture deterioration during chill storage. 2010, 85, 402-9. Tan, C.Q.; Wei, H.K.; Sun, H.Q.; Ao, J.T.; Long, G.; Jiang, S.W.; Peng, J. Effects of Dietary Supplementation of Oregano Essential Oil to Sows on Oxidative Stress Status, Lactation Feed Intake of Sows, and Piglet Performance. Biomed Research International. 2015, 2015, 1-9. Tao, S.; Bubolz, J.; Do Amaral, B.; Thompson, I.; Hayen, M.; Johnson, S.; Dahl, G. Effect of heat stress during the dry period on mammary gland development. J Dairy Sci. 2011, 94, 5976-86. Kim, S.W.; Weaver, A.C.; Shen, Y.B.; Zhao, Y. Improving efficiency of sow productivity: nutrition and health. Journal of animal science and biotechnology. 2013, 4, 26.

Line 126: Piglets were weighed at weaning but weaning data was not provided in table 4.

Response: Thank you for your comment. We apologize for this mistake. Piglets were weaned on day 21 of lactation. We have removed the repeated information. Please seeL128-129, L136, and table 4.

Table 4. Effect of inclusion of resistant starch (RS) or fermented soybean fiber (FSF) in gestation diet on litter performance.

Item

Diet

SEM

p-value

CON1

RS1

FSF1

No. of sows

19

21

22

Average sow parity

4.5

4.8

5.0

No. of pigs per litter

Total piglets born2

15.6

14.0

15.0

0.38

0.24

Piglets born alive2,3

13.3

13.0

13.0

0.33

0.77

Stillbirth2,3

2.0a

0.9b

1.8ab

0.20

0.04

After cross-foster3

12.1

11.6

11.5

0.11

0.12

Pigs weaned3

10.6

10.0

10.3

0.18

0.20

Piglet mean BW, kg

At Birth2,3

1.4

1.4

1.4

0.03

0.99

After cross-foster3

1.6

1.6

1.6

0.03

0.43

On day 7

3.1

3.2

3.1

0.06

0.59

On day 14

5.1

5.1

5.1

0.05

0.67

On day 21

7.0

7.1

7.0

0.07

0.73

Average daily gain, g/d

Day 1 to 7

209.7

220.3

207.0

7.90

0.45

Day 7 to 14

277.9

283.9

274.4

7.54

0.67

Day 14 to 21

268.0

276.4

261.8

11.00

0.65

Day 1 to 21

253.6

265.8

249.5

6.80

0.22

1 CON = control diet group; RS = 5% resistant starch diet group; FSF = 5% fermented soybean fiber diet group.

2 Number of sows in CON, RS and FSF is 22, 21 and 24, respectively.

3 Results were analyzed using the Kruskal-wallis.test.

a,b Means with different superscript letters within a row differ significantly (p < 0.05).

Lines 161 to 169: The blood samples for GLP-1 and peptide YY how were they treated or prepared before the ELISA procedure?

Response: Special thanks to you for your reminder. We have revised it in the meterials and methods about blood sample processing and treatment. Before feeding and at 2 h after feeding on day 70 of gestation, and before feeding on day 109 of gestation, blood samples were collected from 24 sows (eight sows per dietary treatment) by ear vein puncture with a minimum amount of stress into vacuum tubes (5 mL) and labeled heparinized tubes (5 mL). Collection tubes for measuring GLP-1 and PYY blood samples contained a mixture of inhibitors (dipeptidyl peptidase IV inhibitor [10 μL/mL blood], aprotinin [50 μL/mL blood], and Pefabloc SC [50μL/mL blood]) to prevent degradation of the peptides to be measured [1,2]. Samples for serum assays (tubes containing no anticoagulant) were stored at room temperature for 4 h and then centrifuged for 5 min at 5,000 × g at 4 °C to collect the serum, followed by storage in labeled microfuge tubes at −80 °C until further analysis. Plasma samples were obtained by centrifufation of the blood samples in labeled heparinized tubes at 5,000×g for 5 min at 4 °C and then stored at -80 °C until analysis. Before analyzing GLP-1 and PYY, serum was thawed, and an additional protease inhibitor cocktail (final concentration: 1× Sigma-SIGMAFAST[1×] and dipeptidyl peptidase IV inhibitor KR-62436 (0.5 μM), catalog nos. S8820 and K4264, respectively; Sigma-Aldrich, St Louis, Missouri) was added according to published method [1]. The concentrations of GLP-1 and PYY in serum were analyzed using an ultrasensitive pig GLP-1 and PYY enzyme-linked immunosorbent assay (ELISA) kit (CUSABIO, Wuhan, Hubei, China) according to the operation instructions. Please see L113-120, and L170-174.

Reference:

Gibbons C, Caudwell P, Finlayson G, et al. Comparison of postprandial profiles of ghrelin, active GLP-1, and total PYY to meals varying in fat and carbohydrate and their association with hunger and the phases of satiety[J]. The Journal of Clinical Endocrinology & Metabolism, 2013, 98(5): E847-E855. Mochida T, Take K, Maki T, et al. Inhibition of MGAT2 modulates fat‐induced gut peptide release and fat intake in normal mice and ameliorates obesity and diabetes in ob/ob mice fed on a high fat diet[J]. FEBS Open Bio, 2019.

Lines 194 to 195: What is the p-value associated with that statement?

Response: Two batches of in vitro measurement of physio-chemical properities (swelling capacity, viscosity, and water-blinding capacity) were determined with two replicates in each batch. The data were only presented as means that without statistical analysis. Thank you.

Lines 196 to 198: Where is the data to support that result?

Response: Thank you for your comment. The three dietary treatments were shown to have no significant differences (p > 0.05) in their effect on in vitro fermentability of gestation diets. Meanwhile, the 48-h cumulative gas production and the final asymptotic gas volume showed an increased trend (p = 0.07, p = 0.09, respectively) in the RS group versus the CON or FSF group (Table2).

Table 2. Gas production parameters and concentrations of short-chain fatty acids (SCFA) during the in vitro fermentation of enzymatically hydrolyzed residues of the three diets using faecal inocula from gestation sows.

Item

Diet

SEM

p-value

CON 2

RS 2

FSF 2

Fermented fraction

V3

187.17

197.60

169.99

5.60

0.07

VF4

210.41

222.77

197.96

5.08

0.09

(FRD0)5×100

1.86

2.00

2.31

0.16

0.61

K6

0.10

0.08

0.06

0.01

0.65

(T1/2)7

20.75

20.52

20.93

0.24

0.86

Concentrations of SCFA

Acetic acid, mmol/L

20.08

19.65

16.15

1.15

0.39

Propionic acid, mmol/L

9.17

8.57

6.67

0.64

0.30

Butyric acid, mmol/L

1.59

1.35

1.18

1.10

0.32

Iso-butyric acid, mmol/L

0.49

0.30

0.26

0.06

0.31

Valeric acid, mmol/L

1.50

1.09

0.97

0.13

0.22

Iso-valeric acid, mmol/L

0.83

0.46

0.40

0.12

0.33

Total SCFA8, mmol/L

33.66

31.43

25.63

2.07

0.32

1 Data are given as means ± SEM, n = 2 (Number of observations in fermentation).

2 CON = control diet group; RS = 5% resistant starch diet group; FSF = 5% fermented soybean fiber diet group.

3 V, 48-h cumulative gas production.

4 VF, The final asymptotic gas volume (mL/g).

5 FRD0, Initial fractional rate of degradation at t-value=0 (h-1).

6 K, Fractional rate of gas production at a particular time point (h-1).

7 T1/2, Half-life to asymptote (h).

8 Total SCFA= the sum of acetic, propionic, butyric, iso-butyric, valeric and iso-valeric acid.

Line 210: Correct compound name as CuSO4·5H2O but not CuSO4·5H2O

Response: Thanks for your reminder. We have revised all the incorrect compound names. Please see L218-219.

Line 238: Information about sow numbers contradicts what is in table 3.

Response: Thank you for your comment. The number of sows on the day of delivery is different from the number of sows on lactation. Sows were culled because of poor milk during lactation. Therefore, we specifically indicate the number of sows during lactation in Table 3. Please see the number of sows during the experimental period in Supplementary Materials Table 2.

Supplementary Materials Table S2: The number of sows during the experimental period

Item

Diet

CON1

RS1

FSF1

Mating

26

26

26

At day 109 of gestation2

22

21

24

Farrowing

22

21

24

Culled during lactation3

3

0

2

Weaning

19

21

22

1 CON =control diet group; RS=5% resistant starch diet group; FSF=5% fermented soybean fiber diet group.

2 Sows were culled because of foot pain, abortion, or non-pregnant after breeding, etc.

3 Sows were culled because of poor milk.

Line 253: Information about sow numbers contradicts what is in table 4.

Response: Please see our answer to the above comment about Line 238.

Line 250. The sum of piglets born alive and stillborn do not add up to the total piglets born, table 4.

Response: Thank you for your comment. In fact, we also recorded the number of mummy when we counted the total litter size, but it was not included in the number of stillbirth. Please see the following chart.

(1) Raw data on total births, stillbirths, mummy counts and abnormal births.

Sow

No. of  total piglets born

No. of piglets born alive

No. of stillbirth

No. of  mummy

CON-1

13

12

1

0

CON-2

16

13

3

0

CON-3

18

17

1

0

CON-4

10

10

0

0

CON-5

16

15

1

0

CON-6

19

15

3

1

CON-7

19

16

1

2

CON-8

20

15

5

0

CON-9

16

12

3

1

CON-10

12

11

1

0

CON-11

11

11

0

0

CON-12

16

13

2

1

CON-13

16

13

2

1

CON-14

16

16

0

0

CON-15

19

14

5

0

CON-16

16

14

2

0

CON-17

14

13

1

0

CON-18

19

13

4

2

CON-19

17

13

4

0

CON-20

13

10

3

0

CON-21

18

16

1

1

CON-22

10

10

0

0

RS-1

10

10

0

0

RS-2

13

13

0

0

RS-3

15

15

0

0

RS-4

16

16

0

0

RS-5

15

15

0

0

RS-6

13

12

1

0

RS-7

22

20

2

0

RS-8

14

13

0

1

RS-9

18

17

0

1

RS-10

13

11

2

0

RS-11

14

13

1

0

RS-12

17

16

1

0

RS-13

16

16

0

0

RS-14

16

15

1

0

RS-15

11

8

3

0

RS-16

13

10

2

1

RS-17

12

12

0

0

RS-18

10

10

0

0

RS-19

15

11

4

0

RS-20

10

10

0

0

RS-21

12

11

1

0

FSF-1

14

12

2

0

FSF-2

10

8

2

0

FSF-3

16

14

2

0

FSF-4

12

12

0

0

FSF-5

13

12

1

0

FSF-6

15

13

2

0

FSF-7

20

20

0

0

FSF-8

18

14

4

0

FSF-9

16

16

0

0

FSF-10

22

20

1

1

FSF-11

10

10

0

0

FSF-12

19

16

2

1

FSF-13

15

13

2

0

FSF-14

13

13

0

0

FSF-15

12

11

1

0

FSF-16

15

8

5

2

FSF-17

17

12

5

0

FSF-18

12

12

0

0

FSF-19

20

16

4

0

FSF-20

12

12

0

0

FSF-21

14

12

2

0

FSF-22

13

12

1

0

FSF-23

15

15

0

0

FSF-24

17

9

7

1

 (2) The number of piglets born

Item

Diet

CON1

RS1

FSF1

No. of sows

22

21

24

No. of pigs per litter

Total piglets born

15.6

14.0

15.0

Piglets born alive

13.3

13.0

13.0

Stillbirth

2.0

0.9

1.8

Mummy

0.3

0.1

0.2

CON = control diet group; RS = 5% resistant starch diet group; FSF = 5% fermented soybean fiber diet group.

Line 250: Piglets mean BW and litter weight provide the same information and therefore repetitive.

Response: Thank you for your comment. We have deleted the repetitive information of litter weight. Please see table 4.

Table 4. Effect of inclusion of resistant starch (RS) or fermented soybean fiber (FSF) in gestation diet on litter performance

Item

Diet

SEM

p-value

CON1

RS1

FSF1

No. of sows

19

21

22

Average sow parity

4.5

4.8

5.0

No. of pigs per litter

Total piglets born2

15.6

14.0

15.0

0.38

0.24

Piglets born alive2,3

13.3

13.0

13.0

0.33

0.77

Stillbirth2,3

2.0a

0.9b

1.8ab

0.20

0.04

After cross-foster3

12.1

11.6

11.5

0.11

0.12

Pigs weaned3

10.6

10.0

10.3

0.18

0.20

Piglet mean BW, kg

At Birth2,3

1.4

1.4

1.4

0.03

0.99

After cross-foster3

1.6

1.6

1.6

0.03

0.43

On day 7

3.1

3.2

3.1

0.06

0.59

On day 14

5.1

5.1

5.1

0.05

0.67

On day 21

7.0

7.1

7.0

0.07

0.73

Average daily gain, g/d

Day 1 to 7

209.7

220.3

207.0

7.90

0.45

Day 7 to 14

277.9

283.9

274.4

7.54

0.67

Day 14 to 21

268.0

276.4

261.8

11.00

0.65

Day 1 to 21

253.6

265.8

249.5

6.80

0.22

1 CON = control diet group; RS = 5% resistant starch diet group; FSF = 5% fermented soybean fiber diet group.

2 Number of sows in CON, RS and FSF is 22, 21 and 24, respectively.

3 Results were analyzed using the Kruskal-wallis.test.

a,b Means with different superscript letters within a row differ significantly (p < 0.05).

Lines 107 and 148: Blood samples were collected on day 109 of gestation but this contradicts the information on lines 278 and 284.

Response: We are so sorry for our negligence. We have revised them accordingly. Before feeding and at 2 h after feeding on day 70 of gestation, and before feeding on day 109 of gestation, blood samples were collected from 24 sows (eight sows per dietary treatment) by ear vein puncture with a minimum amount of stress into vacuum tubes (5 mL) and labeled heparinized tubes (5 mL). GLP-1 and PYY concentrations were measured using serum from sows before feeding and at 2 h after feeding on day 70 of gestation. Please see L110-113 and L170-171.

Lines 310 to 313; Lines 80 to 82: Depending on which section of the manuscript you read, different objectives are stated.

Response: We apologize for this mistake. We have revised them accordingly. The purpose of this study was to appraise the effects of two available unconventional dietary fiber resources on sows’ reproductive performance through an in vitro-in vivo method to extend their applications in rural livestock production systems. Please see L85-87 and L320-321.

Lines 325 to 326: “Furthermore, including 5% resistant starch fiber in the gestation diet increased the serum concentration of GLP-1”. The baseline values were different for GLP-1. Will the result above be the same if authors had accounted for the baseline values at 2 hour postprandial?

Response: Thank you for your comment. We have rewritten the part based on your suggestions. In the present study, feeding sows with 5% RS increased the serum concentrations of 2-h postprandial GLP-1 and PYY of pregnant sows, which may result in enhanced postprandial satiety. Previous studies have shown that GLP-1 and PYY are important regulators of satiety [1, 2], and they reduced gut motility, delayed gastric emptying, slowed transit time to enhance digestion and nutrient absorption, and thus  reduced appetite,implying that GLP-1 may influence eating at a later meal and hunger in the inter-meal period [3, 4]. Long-term feeding of RS rats continuously up-regulated plasma GLP-1 concentration and large intestinal GLP-1 gene expression within 24 hours [5]. Interestingly, GLP-1 level was also significantly increased before meals in the RS and FSF groups in our results, which might be associated with the fermentation of RS in the lower gut [6]. A previous study also mentioned that microbiota in the distal part of the gastrointestinal tract can ferment dietary fiber to produce SCFAs, especially propionic acid, which could activate the gene expression of free fatty acid receptors in L-cells and promote the generation of GLP-1 and PYY in rodents and humans [7]. Nevertheless, in the present study, inclusion of dietary fibers exhibited no significant effect on fermentability, except for the higher gas production values, probably due to the low dosage of resistant starch. Another possible explanation is the limitations of in vitro enzymatic hydrolysis, which cannot completely mimic the digestive process in the body [8, 9]. In addition, feeding a RS diet was shown to induce satiety by supplying a constant concentration of glucose for a longer time and increasing the concentration of nonestesterified fatty acid in blood in a previous study on gestating sows [10]. We did not measure the levels of these metabolites in the blood, and therefore we are unable to know if glucose and nonestesterified fatty acid in our study would increase as noted by other researcher. Overall, the RS group had the highest levels of satiety hormones before or after a meal, which indicates a reduction in hunger of sows. Please L338-361 in discussion section. Thank you.

Reference:

Schueler, J.L.; Alexander, B.M.; Hart, A.M.; Austin, K.J.; Larson-Meyer, D.E. Presence and Dynamics of Leptin, GLP-1, and PYY in Human Breast Milk at Early Postpartum. Obesity. 2013, 21. Gibbons, C.; Caudwell, P.; Finlayson, G.; Webb, D.-L.; Hellström, P.M.; Näslund, E.; Blundell, J.E. Comparison of postprandial profiles of ghrelin, active GLP-1, and total PYY to meals varying in fat and carbohydrate and their association with hunger and the phases of satiety. The Journal of Clinical Endocrinology & Metabolism. 2013, 98, E847-E55. Blundell, J.E.; Naslund, E. Glucagon-like peptide-1, satiety and appetite control. Br J Nutr. 1999, 81, 259-60. Sleeth, M.L.; Thompson, E.L.; Ford, H.E.; Zac-Varghese, S.E.; Frost, G. Free fatty acid receptor 2 and nutrient sensing: a proposed role for fibre, fermentable carbohydrates and short-chain fatty acids in appetite regulation. Nutrition research reviews. 2010, 23, 135-45. Zhou, J.; Keenan, M.J.; Raggio, A.M.; Tripathy, S.; Shen, L.; McCutcheon, K.L.; Hegsted, M.; Tulley, R.T.; Martin, R.J. Feeding resistant starch maintains elevated plasma levels of GLP-1 and PYY throughout the day and is associated with decreased body fat in rats. Federation of American Societies for Experimental Biology; 2007. Zhou, J.; Keenan, M.J.; Raggio, A.M.; Tripathy, S.; Shen, L.; McCutcheon, K.L.; Hegsted, M.; Tulley, R.T.; Martin, R.J. Feeding resistant starch maintains elevated plasma levels of GLP-1 and PYY throughout the day and is associated with decreased body fat in rats. Federation of American Societies for Experimental Biology; 2007. Byrne, C.; Chambers, E.; Morrison, D.; Frost, G. The role of short chain fatty acids in appetite regulation and energy homeostasis. International journal of obesity. 2015, 39, 1331. Coles, L.; Moughan, P.; Darragh, A. In vitro digestion and fermentation methods, including gas production techniques, as applied to nutritive evaluation of foods in the hindgut of humans and other simple-stomached animals. Animal Feed Science and Technology. 2005, 123, 421-44. Bindelle, J.; Buldgen, A.; Lambotte, D.; Wavreille, J.; Leterme, P. Effect of pig faecal donor and of pig diet composition on in vitro fermentation of sugar beet pulp. Animal feed science and technology. 2007, 132, 212-26. Sapkota, A.; Marchant-Forde, J.N.; Richert, B.T.; Lay, D.C. Including dietary fiber and resistant starch to increase satiety and reduce aggression in gestating sows. Journal of Animal Science. 94, 2117.

Also only one blood sample was collected from 24 fasted sows according to lines 105 to 110. How did authors obtain blood samples before and after feeding?

Response: Special thanks to you for your reminder. We have revised it in the materials and methods about blood sample processing and treatment. Before feeding and at 2 h after feeding on day 70 of gestation, and before feeding on day 109 of gestation, blood samples were collected from 24 sows (eight sows per dietary treatment) by ear vein puncture with a minimum amount of stress into vacuum tubes (5 mL) and labeled heparinized tubes (5 mL). Please see L110-113.

Reviewer 3 Report

LINE                                    COMMENT

L18                 rearrange sentence to de -emphasize swelling capacity effect                          satiety, contribute to greater swelling capacity...sows ; delete                          "due to higher swelling capacity, leading to" ...

L21                 delete" different" ; replace with   " two"

L26                 according to the statistics section, significance was set at 

                       p<0.05 Therefore, 0.09. 0.07 is not significant. It is a 

                        tendency. Rewrite these sentences ,accordingly.

L42                  delete "the" 

L44                  add frequent i.e., to the frequent occurrence

L45                  delete "frequently"

L50                  delete "the"

L64                   delete "be" ; replace with "are"

L64                   delete "They" ; replace with" Resistant starches"

L66                    delete "they" ; replace with "RS"

L71                     add "to diets"  ; starch to diets during

L78                    add comma after satiety

L82                   delete "the"

L93                   place a period after (2012) .     ; delete "and" ; capitalize 

                        Their

L94                   delete hand; replace with " had "

L97                   delete "the"                         

L100                 rewrite to read "on the day of parturition"

 L107                days should be day

L112                  salivary should be saliva   should replace salivary with 

                          saliva throughout text

L124                 When ?  What day range did weaning occur ? 

L153-156           This sentence goes in the discussion section

L157                  delete' an' ; replace with' a'   

L164                  change salivary to saliva

L165-169           This sentence goes in the discussion section

L171                  write out 24---twenty four

L174                  Rewrite sentence.   The number of water drinking episodes 

                           and body position changes were recorded...

L175-177           Rewrite sentence to past tense.

L181-182           This sentence goes in an earlier section.

L194                  replace resistant starch  diet with RS  

L195                  add the----the three

L283-296           change salivary to saliva

L321                  add the--- the diet

L333                   delete studies; add "In a previous study"

L343                 These results agree  ; delete accorded

L350                 replace salivary with saliva

L354                 replace "resulting in" with "possibly contributing to" 

L358                 replace "showed that"  with"have shown"

L357-359           These statements are most important . This narrative should be expanded to address the effects(s) of total fiber in the diet. The NDF  and CF components are similar across treatments. All three diets included WB and SBH such that total fiber feedstuffs in the diets were  similar 23%,30%,25% and the WB and SBH feedstuffs were similar 23%,25%,20% for CON, RS and FSF, respectively. These are all three high fiber diets . Therefore, any conclusions drawn should take this fact into consideration. Three high fiber (HIGHLY DIGESTIBLE FIBER) diets with minimal difference in lowly digestible fiber could be why fewer sign. differences were observed.  In that regard;the RS diet is numerically higher in fiber than the other diets-- what affects could this have had? The authors should address this concern in the discussion.         

... 

Author Response

We would like to thank you for the constructive advice on how to improve the quality of the paper. The following is a point-to-point response to the reviewers’ comments.

Reviewer #3 (Comments for the Author):

L18 rearrange sentence to de-emphasize swelling capacity effect satiety, contribute to greater swelling capacity...sows; delete "due to higher swelling capacity, leading to" ...

Response: Thank you for your comment. We have rewritten the sentence. Results indicated that inclusion of 5% resistant starch with greater swelling capacity in the gestation diet was beneficial to enhancing the postprandial satiety, alleviating the stress status, reducing the abnormal behaviors and thus lowering the stillbirth rate of sows. Please see L18-20.

L21 delete" different" ; replace with   " two"

Response: Thank you for your comment. We have revised it accordingly. Please see L23.

L26 according to the statistics section, significance was set at p<0.05 Therefore, 0.09. 0.07 is not significant. It is a tendency. Rewrite these sentences ,accordingly.

Response: Thank you for your comment. We have revised it accordingly. Results showed that swelling capacity was higher in the RS diet than in the CON or FSF diet. Meanwhile, the 48-h cumulative gas production and the final asymptotic gas volume after in vitro fermentation of gestation diets showed an increased trend (p = 0.07, p = 0.09, respectively) in the RS diet versus the CON or FSF diets. Please see L28-31.

L42 delete "the"

Response: Thank you for your comment. We have revised it accordingly. Please see L47.

L44 add frequent i.e., to the frequent occurrence

Response: Thank you for your comment. We have revised it accordingly. Please see L48-51.

L45 delete "frequently"

Response: Thank you for your comment. We have revised it accordingly. Please see L48-51.

L50 delete "the"

Response: Thank you for your comment. We have revised it accordingly. Please see L53-56.

L64 delete "be" ; replace with "are"

Response: Thank you for your comment. We have revised it accordingly. Please see L69.

L64 delete "They" ; replace with" Resistant starches"

Response: Thank you for your comment. We have revised it accordingly. Please see L69.

L66 delete "they" ; replace with "RS"

Response: Thank you for your comment. We have revised it accordingly. Please see L71.

L71 add "to diets"  ; starch to diets during

Response: Thank you for your comment. We have revised it accordingly. Please see L76.

L78 add comma after satiety

Response: Thank you for your comment. We have revised it accordingly. Please see L83.

L82 delete "the"

Response: Thank you for your comment. We have deleted the word. Please see L87.

L93 place a period after (2012) . ; delete "and" ; capitalize Their

Response: Thank you for your comment. We have revised it accordingly. Please see L98.

L94 delete hand; replace with " had "

Response: Thank you for your comment. We have revised it accordingly. Please see L98.

L97 delete "the"                        

Response: Thank you for your comment. We have rewritten the sentence. Please see L100-102.

L100 rewrite to read "on the day of parturition"

Response: Thank you for your comment. We have revised it accordingly. Please see L105.

L107 days should be day

Response: Thank you for your comment. We have revised it accordingly. Please see L110.

L112 salivary should be saliva should replace salivary with saliva throughout text

Response: Thank you for your comment. We apologize for this mistake. We have revised it and other similar mistakes accordingly. Please see L122, L125, L175, L297, L298, and L304.

L124 When ?  What day range did weaning occur ?

Response: We are sorry for not presenting the information clearly. Piglets were weaned at day 21 of lactation. Please see L128-129, and L136.

L153-156 This sentence goes in the discussion section

Response: Thank you for your comment. We have deleted the sentence and moved this part to discussion.. Please see L370-374.

L157 delete' an' ; replace with' a'  

Response: We are very sorry for the mistake. We have revised it accordingly. Please see L166.

L164 change salivary to saliva

Response: We apologize for this mistake. We have revised it accordingly. Please see L175.

L165-169 This sentence goes in the discussion section

Response: Thank you for your comment. We have deleted the sentence and moved this part to discussion. Please see L348-351.

L171 write out 24---twenty four

Response: Thank you for your comment. We have revised it accordingly. Please see L178.

L174 Rewrite sentence. The number of water drinking episodes and body position changes were recorded...

Response: Thank you for your comment. We have rewritten the sentence based on your suggestions. Please see L182.

L175-177 Rewrite sentence to past tense.

Response: Thank you for your comment. We have rewritten the sentence. Please see L183-186.

L181-182 This sentence goes in an earlier section.

Response: We are sorry for not presenting the information clearly. We have rewritten the sentence. Gas production parameters and fermentation metabolite production were analyzed using the MIXED procedure of SAS (SAS Institute Inc., Cary, NC, USA) according to published methods. Please see L190-191.

L194 replace resistant starch diet with RS 

Response: Thank you for your comment. We have revised it accordingly. Please see L203.

L195 add the----the three

Response: Thank you for your comment. We have revised it accordingly. Please see L204.

L283-296 change salivary to saliva

Response: We apologize for this mistake. We have revised it accordingly. Please see L297-304.

L321 add the--- the diet

Response: Thank you for your comment. We have added the word. Please see L333.

L333 delete studies; add "In a previous study"

Response: Thank you for your comment. We have revised it accordingly. Please see L348.

L343 These results agree; delete accorded

Response: Thank you for your comment. We have revised it accordingly. Please see L364.

L350 replace salivary with saliva

Response: Thank you for your comment. We have rewritten the sentence. Please see L379-380.

L354 replace "resulting in" with "possibly contributing to"

Response: Thank you for your comment. We have revised it accordingly. Please see L385.

L358 replace "showed that" with"have shown"

Response: Thank you for your comment. We have revised it accordingly. Please see L399.

L357-359 These statements are most important . This narrative should be expanded to address the effects(s) of total fiber in the diet. The NDF and CF components are similar across treatments. All three diets included WB and SBH such that total fiber feedstuffs in the diets were similar 23%, 30%, 25% and the WB and SBH feedstuffs were similar 23%, 25%, 20% for CON, RS and FSF, respectively. These are all three high fiber diets. Therefore, any conclusions drawn should take this fact into consideration. Three high fiber (HIGHLY DIGESTIBLE FIBER) diets with minimal difference in lowly digestible fiber could be why fewer sign differences were observed. In that regard, the RS diet is numerically higher in fiber than the other diets-- what affects could this have had? The authors should address this concern in the discussion.

Response: Thank you for your comment. We have rewritten this section based on your suggestions. The diet including 5% RS showed a higher swelling capacity (2.28 ml/g), which means that the total dietary volume for the pregnant sows fed 2.6 kg/d of feed during gestation could reach 5.25 L. A possible explanation for the reduced standing behavior is the increase of postprandial satiety due to the larger dietary volume provided by RS in the feed. Our findings point to the importance that the special physico-chemical (swelling capacity) properties of diets containing RS may have a positive effect on sow reproductive performance regardless of the amount of dietary fiber.

    In the present study, the gestation diets exhibited no effects on sows’ weight and backfat during gestation and lactation, piglet BW, or weaning-to-estrus interval. Several factors may have converged to prevent RS from influencing reproduction performance of sows. Likewise, a previous study found no improvement in sow or litter performance when soybean hulls were offered to sows beginning 2 d postmating to d 109 of gestation [1]. Similar levels of lowly digestible fibers in the three diets of our study could be one of the reasons for fewer differences observed in the reproductive performance. Our research results are not in line with a previous study showing that feeding sows with a high-swelling diet containing konjac flour fiber during gestation increased the subsequent lactation feed intake of sows and tended to improve the performance of piglets [2], but the response is variable [3]. Timing of fiber addition to sow diets might be important. Several researchers began to feed fiber diets at breeding and continued throughout gestation over multiple parities, and reported increases in litter size after 2 or 3 parities [4, 5]. Finally, the average parity of the sows reached the fifth parity in our study, and the excessive age of the sows may also mask the beneficial effects of RS [6]. However, few reports have shown that the high swelling diet with the inclusion of RS during one reproduction cycle could reduce the stillbirth rate. Our findings point to the importance that replacing corn with RS can have a positive effect on sows and thus the potential as unconventional feed ingredients. Please see the dicussion section. Thank you.

References

Renteria, F.J. Effects of soluble and insoluble dietary fiber on diet digestibility and sow performance. 2004. Sun, H.; Zhou, Y.; Tan, C.; Zheng, L.; Peng, J.; Jiang, S. Effects of konjac flour inclusion in gestation diets on the nutrient digestibility, lactation feed intake and reproductive performance of sows. Animal. 2014, 8, 1089-94 Grieshop, C. Nonstarch polysaccharides and oligosaccharides in swine nutrition. Swine nutrition. 2001. 4.Ewan, R.; Crenshaw, J.; Crenshaw, T.; Cromwell, G.; Easter, R.; Nelssen, J.; Miller, E.; Pettigrew, J.; Veum, T. Effect of addition of fiber to gestation diets on reproductive performance of sows. J Anim Sci. 1996, 74, 190. Allee, G.L. Alfalfa haylage for sows during gestation. 1981, 38-41. Holt, J.; Johnston, L.J.; Baidoo, S.K.; Shurson, G.C. Effects of a high-fiber diet and frequent feeding on behavior, reproductive performance, and nutrient digestibility in gestating sows. J Anim Sci. 2006, 84, 946-55.

Round 2

Reviewer 1 Report

Several points rased in my first round have been revised and the manuscript has been improved. Thus, the revised manuscript is now suitable for publication in Animals.